# Health Disparities and Comparison of Psychiatric Medication Use before and after the COVID-19 Pandemic Lockdown among General Practitioner Practices in the North East of England

**DOI:** 10.3390/ijerph20116034

**Published:** 2023-06-02

**Authors:** Ge Yu, Eugene Y. H. Tang, Yu Fu

**Affiliations:** 1NIHR Applied Research Collaboration North East and North Cumbria, Northumberland and Tyne and Wear NHS Foundation Trust, The Clocktower Building, St Nicholas Hospital, Gosforth, Cumbria, Newcastle upon Tyne NE3 3XT, UK; 2Population Health Sciences Institute, Faculty of Medical Sciences, Newcastle University, Baddiley-Clark Building, Richardson Road, Newcastle upon Tyne NE2 4AX, UK; 3Primary Care & Mental Health, University of Liverpool, Liverpool L69 3GL, UK

**Keywords:** mental health services, psychiatric medication, COVID-19, lockdown, real-world evidence, general practitioner practice, health disparities

## Abstract

Background: Psychiatric medications play a vital role in the management of mental health disorders. However, the COVID-19 pandemic and subsequent lockdown limited access to primary care services, leading to an increase in remote assessment and treatment options to maintain social distancing. This study aimed to investigate the impact of the COVID-19 pandemic lockdown on the use of psychiatric medication in primary care settings. Methods: We conducted a retrospective claims-based analysis of anonymized monthly aggregate practice-level data on anxiolytics and hypnotics use from 322 general practitioner (GP) practices in the North East of England, where health disparities are known to be higher. Participants were all residents who took anxiolytics and hypnotics from primary care facilities for two financial years, from 2019/20 to 2020/21. The primary outcome was the volume of Anxiolytics and Hypnotics used as the standardized, average daily quantities (ADQs) per 1000 patients. Based on the OpenPrescribing database, a random-effect model was applied to quantify the change in the level and trend of anxiolytics and hypnotics use after the UK national lockdown in March 2020. Practice characteristics extracted from the Fingertips data were assessed for their association with a reduction in medication use following the lockdown. Results: This study in the North East of England found that GP practices in higher health disparate regions had a lower workload than those in less health disparate areas, potentially due to disparities in healthcare utilization and socioeconomic status. Patients in the region reported higher levels of satisfaction with healthcare services compared to the England average, but there were differences between patients living in higher versus less health disparate areas. The study highlights the need for targeted interventions to address health disparities, particularly in higher health disparate areas. The study also found that psychiatric medication use was significantly more common in residents living in higher health disparate areas. Daily anxiolytics and hypnotics use decreased by 14 items per 1000 patients between the financial years 2019/20 and 2020/21. A further nine items per 1000 decreased for higher health disparate areas during the UK national lockdown. Conclusions: People during the COVID-19 lockdown were associated with an increased risk of unmet psychiatric medication demand, especially for higher health disparate areas that had low-socioeconomic status.

## 1. Introduction

The COVID-19 pandemic and subsequent nationwide lockdown have had a detrimental effect on people’s mental well-being. This has led to a drastic surge in mental health problems, affecting both previously healthy individuals and those with pre-existing mental health conditions [1,2,3]. The impact of this surge has been quite substantial, with roughly 10 million people in England necessitating new or additional mental health assistance, accounting for approximately 20% of the population [4]. People with pre-existing mental health conditions were especially vulnerable during this period, experiencing exacerbated symptoms, difficulties accessing necessary services and support, and an increased risk of relapse and suicidal behavior [1]. As a consequence, there has been a higher demand for crucial mental health services and treatment, exceeding the capacity of these services.

The pandemic has also brought about significant disruptions to the usual practices of mental health services not only in the United Kingdom (UK) but also globally [5,6]. The UK government initiated national restrictions on 23 March 2020 to control the infection rate, forestall the National Health Service (NHS) from becoming overwhelmed, and lessen the number of COVID-19-associated fatalities. Nonetheless, recent studies have indicated that the adaptations necessary to provide mental healthcare services during this prolonged period of infection-control measures may have had an inequitable impact on individuals with pre-existing mental health conditions (e.g., autism [7], obsessive compulsive disorder [8], and substance use disorder [9]).

Furthermore, the challenges of attending in-person review appointments and the closure of support services are likely to have had an impact on all individuals who were receiving or in need of active mental health treatment [10]. For instance, initiating new medications with patients who are being seen virtually can be challenging, which may also impede their use. These challenges may have further worsened and deepened the existing structural health inequalities that were already being targeted by quality improvement initiatives before COVID-19 [11]. Patients in higher health disparate areas may have experienced limited access to telehealth services during the pandemic, resulting in a reduced availability of psychiatric medications for them. Patients’ changing attitudes have also contributed to these challenges, as the fear of contracting the virus coupled with national lockdowns has led to a decrease in people seeking medical attention, especially in higher health disparate areas where individuals already faced significant challenges in accessing mental health services due to issues such as poverty, limited resources, and stigma.

Psychiatric medications, such as anxiolytics and hypnotics, are frequently prescribed to alleviate symptoms of anxiety, depression, and insomnia. Recent studies have reported a rise in prescriptions for psychiatric medication, which appears to coincide with an increase in psychological symptoms during the pandemic when compared to the preceding year [12,13,14]. However, it is not well understood how the pandemic and the restrictions introduced to contain the infection spread have affected the use of psychiatric medication, particularly concerning those from higher health disparate areas. Given the widespread use of these medications and their potential impact on mental health outcomes, it is crucial to gather information to develop effective approaches to addressing the mental health needs of these communities. Therefore, in this study, we aim to examine the association between trends in psychiatric medication use among general practitioner (GP) practices located in regions with higher health disparity and the COVID-19 pandemic lockdown. This inquiry is important because learning from healthcare service changes is vital to understand whether and to what extent the pandemic and subsequent lockdown influenced medication use for individuals with the most significant needs, informing proactive planning for future pandemics.

## 2. Materials and Methods

We performed a retrospective cohort study at the practice level and conducted a multivariate analysis of psychiatric medication use from April 2019 to March 2021 using publicly available datasets.

### 2.1. Setting

This study was carried out in the North East of England, which has the lowest healthy life expectancy, with an average of 59 years for men and 61 years for women, compared to a national average of 64 years for men and 65 years for women [15,16]. Furthermore, previous research indicates that the North East region exhibits higher health disparities compared to the rest of England. Specifically, the region has elevated mortality rates from cancer, circulatory diseases, and respiratory diseases, as well as higher rates of hospital admissions for alcohol-related conditions and self-harm [17,18]. The findings from this study could aid in comprehending the impact of COVID-19 on higher health disparate areas in England and help direct the development of the region’s healthcare strategies.

### 2.2. Time Periods of the COVID-19 Lockdown in England

The time periods [19] studied were

Pre-lockdown (23 March 2019–22 March 2020);First national lockdown (23 March–3 July 2020);Minimal lockdown restrictions (4 July–13 September 2020);Reimposing restrictions (14 September–4 November 2020);Second and third lockdowns (Lockdowns 2 and 3 were combined because of a short interval between them) (5 November–7 March 2021).

For our regression analysis, we defined the post-lockdown period as the last four periods.

### 2.3. Medication Use Data

The OpenPrescribing database (OpenPrescibing.net, Bennett Institute for Applied Data Science, University of Oxford, 2022) was used to collect prescription information in the North East of England between April 2019 and March 2021. The database is derived from claims data for prescriptions that community pharmacies submit for reimbursement from the NHS in England. It contains anonymized data on medicines prescribed as prescription items, quantity and cost incurred. The data encompasses prescriptions written by GPs and other non-medical prescribers who are attached to practices, but it does not include private prescriptions.

The primary focus of this study was on the usage of anxiolytics and hypnotics, with the volume of their standardized average daily quantities (ADQs) per 1000 patients serving as the primary outcome measure. The use of standardized ADQs facilitates comparisons between different medications or populations by accounting for the size of the population being analyzed [20]. This approach can help detect patterns and trends in medication use and inform decision-making regarding resource allocation, formulary management, and public health planning.

The ADQ is calculated by dividing the total number of dispensed doses of a specific medication by the number of patients who take that medication and then multiplying it by the number of days for which the medication was dispensed. The resulting value is then standardized by dividing it by 1000 to account for differences in the size of the population being studied. Monthly practice-level ADQ of anxiolytics and hypnotics data were downloaded for the study period.

### 2.4. GP Practice Characteristics

We obtained GP practice characteristics indicators for all practices in the North East of England from the Fingertips data (2020/21) to identify population characteristics attributed to each GP practice that may contribute to differences in psychiatric medication use. Fingertips data is a public health data analysis and visualization tool developed by Public Health England [21]. It provides access to a wide range of health and well-being indicators at national, regional, and local levels in England.

To assess the influence of socioeconomic factors on psychiatric medication use, we used the index of multiple deprivations (IMD) 2019 provided for geographic areas (lower-level super output areas or LSOAs). LSOAs are geographic areas used in the United Kingdom for statistical purposes, and they are smaller than local authorities but larger than individual postcodes. The IMD measures deprivation across dimensions such as income, employment, education, health, crime, access to housing and services, and living environment. The higher value of the IMD score is indicative of a greater degree of area-based deprivation in comparison to other areas, which is related to spatial inequalities in health. We also used proportions of registered patients aged 75 or over in each practice as a proxy measure for the intensity of GP practice workload, with older people accounting for a relatively large proportion of GP consultations and prescriptions [22].

To evaluate the patient experience of and access to their GP services, we employed measures such as the proportion of patients reporting positive experiences, satisfaction with phone access and practice appointment times, and overall experience of making an appointment. We also examined the demand for GP services through long-term health conditions prevalence rates, as people with long-term physical health conditions commonly experience mental health problems and have high healthcare needs (50% of all GP appointments) [23]. Lifestyle indicators, such as the percentage of patients with caring responsibility, those in paid work or full-time education, and those who are unemployed, were also taken into account.

Finally, mental health indicators were assessed, including the quality and outcomes framework (QOF) prevalence of mental health and the percentage of patients with a new diagnosis of depression. By considering these various factors, we provided a picture of the characteristics of these GP practices and their patients that may impact the use of psychiatric medications in the North East of England.

### 2.5. Statistical Methods

A descriptive analysis was performed to examine potential regional disparities in these prescribing GP practices. The region-level data was compared with the national average in England to provide a broader context and enhance the comprehension of medication usage in the North East region. This comparison can be useful for policymakers and healthcare providers who seek valuable information for decision-making purposes. Temporal trends in the monthly ADQs of anxiolytics and hypnotics use in practice before and after the onset of the COVID lockdown are illustrated in Figure 1.

The IMD score for 2019 was divided into quartiles prior to conducting the multivariable regression model analysis in order to enable easy interpretation of the results. A random-effects model was used to examine the relationship between the COVID-19 lockdown and the use of anxiolytics and hypnotics. We included calendar months as covariates to account for seasonal variations in the data. Specifically, we added 11 binary variables to represent each month, with one month (April) serving as the reference category. The significance of each month suggests that there may be certain months that have a significant impact on anxiolytics and hypnotics use, even after accounting for the effect of the lockdown. To account for pre-existing differences across practices, the model included a range of baseline practice characteristic covariates, as listed previously. In order to minimize bias resulting from unobserved time-invariant factors that were correlated with the use of anxiolytics and hypnotics, we estimated models with random practice effects. The random effects specification was assumed to be valid as it is more efficient than the fixed effects specification and makes use of both within- and between-practice variation in the data, given that the time-varying explanators are uncorrelated with unobserved time-invariant practice factors.

Data statistics were presented as mean, standard deviation (SD), median, and range. *p* < 0.05 was considered to be statistically significant. All analyses were conducted using Stata 17 (StataCorp LLC, College Station, TX, USA). We reported robust standard errors clustered at the practice level.

## 3. Results

### 3.1. Health Disparities

The study collected monthly data on anxiolytics and hypnotics use from 322 GP practices in the North East of England over two financial years from 2019/20 to 2020/21, resulting in 7728 practice-month observations.

The study compared the characteristics of these practices to England’s average (as displayed in Table 1). It was found that GPs in the North East of England had a heavier workload than England’s average, with a higher proportion of elderly patients (8.7% vs. 7.9%). Furthermore, within the North East of England, GP practices located in higher health disparate areas exhibited a lower workload on average than those in less health disparate areas, with a significantly lower proportion of patients aged over 75 (7.6% vs. 9.8%).

Patients in the North East of England reported higher levels of satisfaction with healthcare services provided by their local GP practices compared to the average in England. Specifically, patients reported higher satisfaction rates in terms of the quality of care received (86.8% vs. 83.0%), the experience of phone access (75.6% vs. 67.6%), appointment times (67.3% vs. 62.7%), and making appointments (75.6% vs. 70.6%). However, patients living in higher health disparate areas were found to be less satisfied with their GP practices (85.4% vs. 88.2%), phone access (73.4% vs. 77.8%), appointment times (66.6% vs. 68.0%), and making appointments (73.9% vs. 77.3%), compared to those living in less health disparate areas. In the North East of England, the prevalence of long-term conditions among registered patients in GP practices exceeded the English average, with a percentage of more than 5% (56.3% vs. 51.1%). Patients living in higher health disparate areas were found to have a higher prevalence of long-term conditions compared to those living in less health disparate areas (57.4% vs. 55.2%).

There was a negligible contrast in the percentage of individuals in higher health disparate areas versus less health disparate areas who reported responsibilities for care (19.9% vs. 20.0%). However, it was observed that a larger proportion of patients in the North East of England had the responsibility of caring for someone compared to the English average (20.0% vs. 18.2%).

A lower percentage of individuals in the North East of England were engaged in paid work or full-time education compared to England’s average (57.0%% vs. 61.8%). Conversely, a slightly higher percentage of people in the region were unemployed compared to England’s average (6.2% vs. 5.5%). Additionally, the data show that in higher health disparate areas, there were lower proportions of people registered with practices who were engaged in paid work or full-time education compared to less health disparate areas (with a difference of 2.6%). Conversely, there were higher proportions of unemployed individuals in higher health disparate areas compared to less health disparate areas (with a difference of 4.9%).

The data reveal that there was no significant difference in the reported prevalence of mental illness in the QOF between the North East of England and England average. However, the proportion of individuals who received a new diagnosis of depression in the North East of England was slightly higher than England’s average, with a figure of 1.5% compared to 1.4% in the whole of England. Moreover, the QOF reported prevalence of mental illness in higher health disparate areas was 1.1% compared to 0.9% in less health disparate areas. The proportion of individuals who received a new diagnosis of depression was higher in higher health disparate areas at 1.6% compared to 1.4% in less health disparate areas.

### 3.2. Trends in Psychiatric Medication Use

Our observation revealed a monthly variation during the 2019/20 financial year, with the highest medication usage in the months of March and April and consistently lowest numbers in January and February. The first UK lockdown that commenced on 23 March 2020, which required people to stay at home, resulted in a sharp drop in ADQs of medication use. Although the volumes were gradually recovering as restrictions eased, the ADQs remained below pre-lockdown levels until the end of the study period. The second lockdown, which commenced on 5 November 2020, was associated with a further reduction in the monthly ADOs of prescriptions and a bigger gap in the same month between the 2019/20 and 2020/21 financial years compared to the first lockdown. However, no wider gap was found from the second lockdown to the third lockdown, which began in January 2021. The partial removal of restrictions associated with each lockdown, as indicated by the green dashed lines, resulted in an immediate increase in medication use. We noted a peak increase when all non-essential retailers were allowed to reopen in England from 15 June 2020 and a second peak increase when the second lockdown ended on 2 December 2020.

### 3.3. The Impact of COVID Lockdown

Table 2 presents the adjusted regression coefficients that exhibit the relationship between anxiolytics and hypnotics use and the pandemic lockdown, the interaction between socioeconomic status and the lockdown, seasonal variation, and GP practice characteristics. The lockdown was found to have a significant negative association with the volume of anxiolytics and hypnotics used. From April 2020 to March 2021, there was an average of 14 fewer ADQs (*p* < 0.001) compared to the 2019/20 financial year average for the same period. Furthermore, on average, 50 more ADQs of anxiolytics and hypnotics (*p* = 0.011) were used in GP practices serving higher health disparate populations compared to those serving less health disparate populations.

We also observed a noteworthy and adverse interaction effect between higher health disparities and pandemic lockdown on anxiolytics and hypnotics use (β = −9.09, *p* = 0.007), suggesting that patients living in higher health disparate areas had even more limited access to these medications during the lockdown than those in less health disparate areas. This may be due to a variety of factors, such as reduced access to healthcare services, financial constraints, or greater levels of stress and anxiety related to the pandemic. The adverse effect highlights the unequal effect of the pandemic lockdown, which may worsen the already-existing health inequalities in mental health service utilization among higher health disparate areas.

Monthly variation was seen in daily medication use, with the highest medication usage in April and consistently lowest numbers in February, representing a reduction of 31 items per 1000 per day compared to April. In addition, we found that practices with a higher workload, as indicated by a greater proportion of patients aged over 75 years, had a higher prescription volume, with an average of over 14 prescriptions more per 1000 per day. Medication use volume also appeared to be based on the morbidity needs of the practice population, with higher medication use volumes in practices reporting a higher prevalence of long-term health conditions.

## 4. Discussion

### 4.1. Health Disparities

This study conducted in the North East of England revealed important findings regarding healthcare services and socioeconomic inequalities in health. Firstly, the workload of GP practices in the region was higher than the average workload in England, which is in line with previous studies [24,25]. However, within the region, GP practices in higher health disparate areas had a lower workload as compared to those in less health disparate areas. This observation could be attributed to the fact that patients living in less health disparate areas are more willing to access healthcare. Extensive research supports the strong relationship between socioeconomic status, healthcare utilization, and healthy aging [26,27,28,29,30,31]. These disparities may impact equitable access to high-quality healthcare services. Secondly, patients in the region reported higher levels of satisfaction with their healthcare services than the England average. This may be because people in the North East of England have low expectations of the healthcare system, or they are grateful for any care that they are able to receive, despite the challenges they face in accessing care [32,33]. Within the region, there were differences among patients living in higher health disparate areas versus less health disparate areas. The findings suggest that further efforts may be necessary to raise people’s expectations of healthcare quality to ensure equitable access to high-quality healthcare services for patients living in higher health disparate areas. Thirdly, there may be a greater need for healthcare services that focus on managing long-term conditions in the North East of England, particularly in higher health disparate areas. As such, it may be beneficial to implement targeted interventions aimed at improving the health outcomes of patients living with long-term conditions in these areas. Fourthly, disparities in paid work and full-time education were observed between higher and less health disparate areas. This disparity can have negative consequences on overall health and well-being, as employment and education are important determinants of health. This highlights the need for targeted interventions and policies to address these disparities and promote equal access to employment and education opportunities, particularly in higher health disparate areas. The study also highlighted higher rates of mental health issues in higher health disparate areas, indicating the need for improving access to mental health services and resources in these areas. It is also important to increase awareness and understanding of mental health issues in these areas and to provide support and education to individuals who may be at risk. Additionally, efforts should be made to address the root causes of mental health disparities, such as poverty and social inequality.

### 4.2. Psychiatric Medication Use

This study found a substantial drop in psychiatric medication use following the lockdown, particularly in higher health disparate areas. This is consistent with previous studies that have shown that disadvantaged individuals are more likely to experience barriers to accessing mental health care and are less likely to receive appropriate treatment for their mental health conditions [34,35]. The pandemic and subsequent lockdown may have exacerbated these existing disparities by limiting access to mental health services and increasing stressors such as job loss, financial instability, and social isolation, which can negatively impact mental health [6,36].

The study also highlighted that during the pandemic, patients may not be able to access psychiatric medications, which can impact the continuity of care for them in managing their mental health. Proper dose titration is crucial for individuals taking psychiatric medication, as abruptly stopping or restarting a high-dose prescription can lead to dangerous side effects and withdrawal symptoms [37]. The disruption of medication access during the pandemic may have contributed to medication nonadherence and treatment discontinuation, which can increase the risk of relapse and worsen mental health outcomes.

In sum, the study’s findings align with previous research in the UK and internationally into the uneven regional impact of the pandemic and its implications for socioeconomic inequalities in health. They provide preliminary support to the concerns that the pandemic and subsequent lockdown have increased inequality in healthcare [11,38], particularly in higher health disparate areas [39,40]. These findings have important implications for future crises, and there is a need to assess the strengths and weaknesses of healthcare services to better plan and support mental health in disadvantaged communities. Targeting resources, such as introducing new models of mental healthcare and increasing the provision of mental health services, may help to alleviate some of the health consequences of the pandemic in vulnerable areas.

## 5. Strengths and Limitations

This study provides valuable insights into the impact of the COVID-19 pandemic and subsequent lockdown on psychiatric medication use, particularly in higher health disparate areas. The use of a random-effect model enabled us to account for medication use patterns that are associated with the local GP practice of residence. The study also utilized high-quality medication use data sourced from pharmacy claims, which ensured a high level of completeness and eliminated the possibility of recall bias. The large sample size and large effect size obtained in this study resulted in a high level of statistical significance in many of the observed associations.

However, it is important to acknowledge the limitations of the study. One of the major limitations is the lack of patient socioeconomic and demographic data included in the dataset. The numbers of medication use are given as aggregate data per GP practice for each month, and thus, potential biases relating to patient socioeconomics and demographics could not be controlled for. Additionally, there is a lack of data on the diagnosis of depression, meaning that some of the medication use may not be for mental illness. Finally, the unexplained variance in the model could be related to other individual and practice changes that occurred during the study period.

To address these limitations and to further explore the impact of the pandemic on mental health, future analyses could integrate multiple individuals and practice time-varying variables. These variables could help to control for potential biases relating to patient demographics and provide a more nuanced understanding of the impact of the pandemic on mental health in higher health disparate areas. Despite these limitations, the findings of this study are significant and provide preliminary support for concerns that inequality in healthcare increased during the pandemic lockdown, particularly in higher health disparate areas. As such, the study highlights the need for targeted interventions and increased provision of mental health services in vulnerable areas to help alleviate some of the health consequences of the pandemic.

## 6. Conclusions

Our study has provided important insights into the impact of the COVID-19 pandemic and subsequent lockdown on mental health service utilizations in higher health disparate areas. Our findings demonstrate a decrease in psychiatric medication use during the period of national lockdown, particularly among those living in higher health disparate areas. These results suggest that the pandemic has exacerbated existing socioeconomic inequalities in healthcare service use, highlighting the need for targeted support in these areas. Our study underscores the importance of addressing the regional disparities in the long-term impacts of COVID-19-related changes on mental health service providers, who need to ensure that patients in higher health disparate areas receive adequate support during and after the pandemic. However, we recognized that future research should delve deeper into the underlying reasons for this decline in psychiatric medication use and identify potential strategies to address this issue. In conclusion, our study emphasized the urgent need for targeted interventions to support mental health services in higher health disparate areas during times of crisis.

## Figures and Tables

**Figure 1 ijerph-20-06034-f001:**
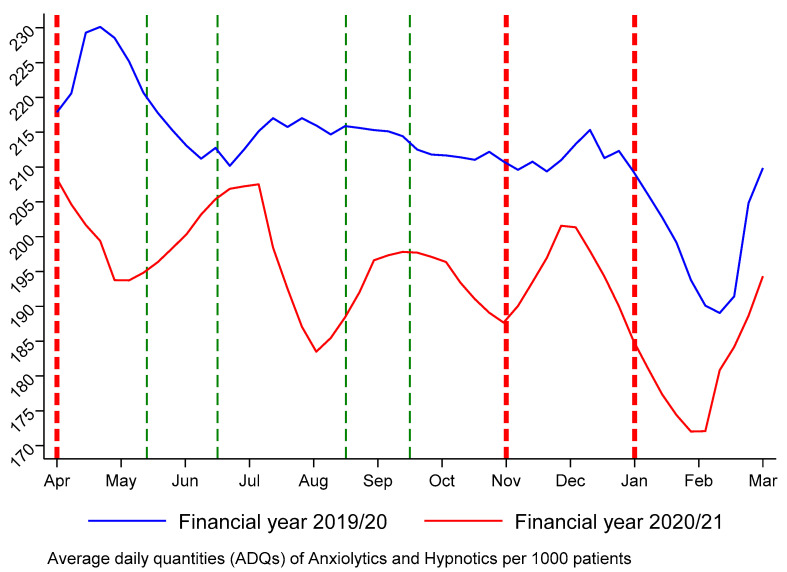
Trends in monthly average daily quantities of anxiolytics and hypnotics prescription in the North East of England.

**Table 1 ijerph-20-06034-t001:** Descriptive statistics of GP practice characteristics in 2020 (n = 322).

	North East England					England	*p*-Value				
	Mean	SD	Median	Range		Mean					
Proportion of patients aged 75 or above	8.72	2.65	8.69	0.07	15.58	7.92	<0.001				
Proportion of reporting a positive experience	86.81	7.65	88.09	54.45	100.00	82.97	<0.001				
Proportion of satisfaction with phone access	75.59	17.98	79.92	20.48	100.00	67.61	<0.001				
Proportion of satisfaction with practice appointment times	67.26	11.30	67.09	31.18	94.43	62.66	<0.001				
Proportion reporting a good overall experience of making an appointment	75.60	11.71	76.58	29.29	99.21	70.64	<0.001				
Prevalence of long-term health conditions	56.31	7.38	56.64	36.77	76.76	51.10	<0.001				
Percentage of caring responsibility	19.96	5.03	20.23	3.00	32.24	18.20	<0.001				
Percentage of reporting to be in paid work or in full-time education	57.03	7.45	56.95	20.78	81.35	61.82	<0.001				
Percentage of reporting to be unemployed	6.16	5.39	5.39	0.00	54.63	5.52	<0.001				
QOF prevalence of mental illness (all ages)	0.96	0.27	0.94	0.12	1.93	0.95	0.085				
Proportion of patients with a new diagnosis of depression	1.51	0.87	1.44	0.07	4.80	1.39	<0.001				
Proportion with a reviewed dementia care plan	41.83	28.34	37.50	0.00	100.00	39.66	<0.001				
Personalized care adjustment (PCA) rate	6.67	8.88	3.30	0.00	54.92	6.49	0.073				
Proportion of patients with severe mental health issues having a comprehensive care plan	40.54	27.03	40.26	0.00	100.00	43.13	<0.001				
	**North East England: two IMD quartiles with the most deprivation**					**North East England: two IMD quartiles with the least deprivation**					** *p* ** **-value**
	**Mean**	**SD**	**Median**	**Range**		**Mean**	**SD**	**Median**	**Range**		
Proportion of patients aged 75 or above	7.61	2.15	7.87	0.07	12.74	9.84	2.63	9.71	0.38	15.58	<0.001
Proportion reporting a positive experience	85.42	7.17	86.41	62.12	98.26	88.19	7.89	90.29	54.45	100.00	<0.001
Proportion of satisfaction with phone access	73.36	18.28	77.11	20.48	98.37	77.83	17.46	82.36	28.03	100.00	<0.001
Proportion of satisfaction with practice appointment times	66.56	10.64	66.50	40.28	88.70	67.95	11.92	67.45	31.18	94.43	<0.001
Proportion reporting a good overall experience of making an appointment	73.93	11.38	75.05	29.29	96.18	77.27	11.84	79.46	43.26	99.21	<0.001
Prevalence of long-term health conditions	57.39	7.53	57.10	37.46	76.76	55.23	7.09	56.13	36.77	72.36	<0.001
Percentage with care responsibilities	19.88	5.45	20.30	3.00	32.24	20.04	4.58	20.13	5.45	31.29	0.157
Percentage reporting to be in paid work or in full-time education	55.73	7.26	56.00	20.78	72.65	58.34	7.42	57.75	37.44	81.35	<0.001
Percentage reporting to be unemployed	8.60	6.32	7.29	0.00	54.63	3.73	2.52	3.07	0.00	10.49	<0.001
QOF prevalence of mental illness (all ages)	1.06	0.25	1.03	0.21	1.93	0.85	0.23	0.84	0.12	1.51	<0.001
Proportion of patients with a new diagnosis of depression	1.60	0.98	1.57	0.12	4.80	1.41	0.73	1.38	0.07	4.10	<0.001
Proportion with a reviewed dementia care plan	44.01	28.68	44.17	0.00	95.12	39.66	27.91	35.09	0.00	100.00	<0.001
Personalized care adjustment (PCA) rate	6.94	9.49	3.33	0.00	54.92	6.40	8.24	3.27	0.00	39.94	0.007
Proportion of patients with severe mental health issues having a comprehensive care plan	44.10	26.37	43.96	0.00	96.43	36.99	27.28	34.50	0.00	100.00	<0.001

IMD: Index of Multiple Deprivation; QOF: Quality and Outcomes Framework.

**Table 2 ijerph-20-06034-t002:** Multivariate regression analysis to describe associations between anxiolytics and hypnotics use volume and predictor variables.

Variables	Coefficient	Standard Error	*p*-Value
**Lockdown**			
Pre-lockdown (financial year 2019/20)	Reference		
Post-lockdown (financial year 2020/21)	−14.06	1.32	<0.001
**IMD 2019 score quartile**			
Less health disparate areas (measured by two IMD quartiles with the least deprivation)	Reference		
Higher health disparate areas (measured by two IMD quartiles with the most deprivation)	50.21	19.65	0.011
**Interaction**			
Interaction between lockdown and higher health disparities	−9.09	3.34	0.007
**Month**			
April	Reference		
May	−2.52	1.27	0.047
June	−9.53	1.19	<0.001
July	−2.77	1.27	0.029
August	−10.40	1.34	<0.001
September	−11.16	1.31	<0.001
October	−5.22	1.30	<0.001
November	−16.70	1.45	<0.001
December	−6.56	1.61	<0.001
January	−12.04	1.85	<0.001
February	−31.24	2.08	<0.001
March	−10.96	1.75	<0.001
**GP practice profile**			
Proportion of patients aged 75 or above	14.37	4.91	0.003
Proportion reporting a positive experience	−0.86	1.71	0.615
Proportion of satisfaction with phone access	0.26	0.72	0.721
Proportion of satisfaction with practice appointment times	2.24	1.30	0.085
Proportion reporting a good overall experience of making an appointment	−1.55	1.52	0.309
Prevalence of long-term health conditions	2.67	1.34	0.046
Percentage with care responsibilities	−2.01	1.42	0.158
Percentage reporting to be in paid work or in full-time education	2.22	1.45	0.124
Percentage reporting to be unemployed	1.22	2.42	0.613
QOF prevalence of mental illness (all ages)	51.02	28.59	0.074
Proportion of patients with a new diagnosis of depression	3.24	9.32	0.728

IMD: Index of Multiple Deprivation: the larger the score, the higher health disparities in the area.

## Data Availability

All data are fully available without restriction.

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
