# Peer review of "Health Disparities and Comparison of Psychiatric Medication Use before and after the COVID-19 Pandemic Lockdown among General Practitioner Practices in the North East of England"

_ijerph, 2023, doi:10.3390/ijerph20116034_

Round 1

Reviewer 1 Report

   Dear authors, 

Thank you for the opportunity to review this manuscript. Here are my comments:

Introduction:

I have no major remark for the Introduction Section. Still, I suggest you to re-arrange the ideas from the paragraph 68-66 and 67-72.

Material and methods:

To complete at the "Statistical methods" subchapter with the other used statistical methods (Anova, predictor variable, s.a).

Results:

No major remarks, only a suggestion referring to a greater clarity of the ideas.

In paragraph 240-242, to specify the year (April and February).

Discussion

Entire re-construction of this chapter to compare the existing results with data from other studies.

Conclusion

More specified conclusions based on results.

Reviewer 2 Report

The manuscript titled “Comparison of psychiatric medication use before and after the COVID-19 pandemic lockdown among general practitioner  practices in disadvantaged communities in England” presents a very timely topic after an unprecedented Pandemic.

This paper needs a major revision before it can be considered as accepted. Below are my comments

Major comments:

1. The result sections are focusing health disparity and use of psychiatric medications between pre and post lockdown period, but the title is only focusing on psychiatric medication use. I suggest the modify the title and add the word “health disparity” . I would remove “disadvantaged communities” words from the title.

Abstract

2. Lines 13-17: The authors should talk about why psychiatric medications are important ? Consider reducing the first sentence and making it simple!

3. Lines 18-25: Remove “most deprived communities” rather say where health disparities are higher and clearly mention “North east of England”. Also, mention how do you defined higher health disparities in NE of England. Add the name of the Rx dataset that was used.

4. Lines 27-30: Add more results here. The manuscript has really good results which need to be highlighted in the abstract section. Add a sentence on health disparity as well

Remove “deprived communities” from the keywords

Introduction

5. First two paragraphs are fine to set stage about the pandemic, but the authors need to add a paragraph regarding psychiatric/anxiolytic and hypnotic medications definition, use and why it is worth of exploring it during this pandemic

6. Lines 58-72: Trim and revise two paragraphs into one paragraph

Materials and Methods

7. Lines 87-90: Provide some stats on lowest healthy life expectancy and higher health inequalities about NE England compared to England with relevant citations

8. In the analysis the authors used only two periods: pre- and post-lockdown, I suggest to define two main periods in the method

9. Replaced the word “deprived” to “Higher health disparate area”. Add a spatial map in the methods section, identifying the both areas with IMD index

10. The authors did not present any demographic and socio-economic data for the dataset, so it would be unjustified naming it deprived communities in NE

11. Add a supplementary file which medications the authors used in the analysis using NDC codes? Name? Provide the whole list of the medications

Results

12. I suggest to create subsections grouped by relevant results For example  

Lines 161-181 can be sub-titled as “General Characteristics of the GP practice in NE England”

Create at least 3 subsections with titled

13. Add sample size in the Table 1 for each group. It is not clear that the authors also calculated the descriptive for England region or not. I suggest to keep the same style for two groups. Such as provide Mean/Median[range] for each region

14. Lines 209-223: Shorten this paragraph. Some parts will go to methods and discussion section

15. Lines 233- 234: Add more justification regarding the adverse interactions effect?

16 Table 2: Define what is the time duration for pre- and post-lockdown as a footnote

Discussion

17 . I suggest to divide the discussion section as same paragraphs like results section. First paragraph of the results should tell the audience what are the main findings of this study

Then provide sufficient discussion on each finding with relevant references

Minor:

1.  Line 78: Remove “disadvantaged community” use Region with higher health disparity

2. Line 83: is it a retrospective/cross-sectional study?

3. Typo in Line 98

4. Line 119? Define Monthly practice-level medication data

5. Line 126: Define LSOA

Reviewer 3 Report

The manuscript, “Comparison of psychiatric medication use before and after the 2 COVID-19 pandemic lockdown” characterizes psychiatric medication prescription in North East England and provides a comparison of this region against the rest of England. The topic area is both interesting and relevant. However, there are some areas of improvement. Chiefly, the manuscript could benefit from a clearer setup of its methodology and presentation of analyses throughout the manuscript. This would include providing more motivation for analyses as some of them, by appearing or being reported first in the results without any prior description seem rather ad hoc or exploratory. Furthermore, the Discussion could benefit from a more detailed integration of specific findings from regression analyses. Other suggestions are provided below.

Abstract

Line 13 - As lockdowns are no longer happening, limits to access as a result of these lockdowns may be referred to in the past tense.

Line 20 - The sentence “participants are...” could also be in the past tense to match tense of other study procedures.

Introduction

What are findings from other studies on the impact of psychiatric and non-psychiatric medication prescription during Covid?

The last paragraph of the introduction 

Methods

What regions constitute North East England? For instance, how many counties are included or how big an area is this? For readers unfamiliar with the UK, what reasons underlie the health inequities and lower life expectancies of this region?

Please briefly describe Fingertips data.

The paragraph “GP practice characteristics” includes factors pertaining to practices, their patients, and their communities; these may be better separated out/ not listed as a single factor. Additionally, to make it easier for readers to keep track of study variables, it is recommended that the baseline characteristics be described in more of a list rather than a single runon paragraph.   

Although the methods list 5 time points, only two are actually used in analyses. While there is a footnote later on in the paper, it would be much clearer to mention that these time periods were collapsed to pre and post in the Methods. 

Results

Comparisons between practices and people in NE England and the rest of England are not previously described in the introduction or in the statistical analyses. So, they seem to come out of nowhere when presented in the Results. This should be added to the relevant sections of the manuscript.

Lines 167 & 174 - How were deprived communities determined?

Line 178: This statement is very unclear - “In the North East of England, more than 5% of the registered patients in GP practices had a long-term condition compared to the England average (56.3% vs 51.1%).”

Figure 1: The Methods describe including the period till March 2021; however, Figure 1 ends at March 2020. What were prescription trends like in the excluded year?

What was the motivation for examining the interaction between lockdown and deprivation? This is not described anywhere in the paper. What do the findings indicate/mean?

What was the motivation for including calendar months as covariates in the regression analyses? Again, this is not included in the Methods. What does it mean that each month is significant (particularly above the effect of pre/post lockdown)?
